{:.logo} PLOS ONE

# Downscaling satellite soil moisture using geomorphometry and machine learning

**Mario Guevara, Rodrigo Vargas** *

University of Delaware, Department of Plant and Soil Sciences, Newark, DE

* rvargas@udel.edu

## Abstract

Annual soil moisture estimates are useful to characterize trends in the climate system, in the capacity of soils to retain water and for predicting land and atmosphere interactions. The main source of soil moisture spatial information across large areas (e.g., continents) is satellite-based microwave remote sensing. However, satellite soil moisture datasets have coarse spatial resolution (e.g., 25–50 km grids); and large areas from regional-to-global scales have spatial information gaps. We provide an alternative approach to predict soil moisture spatial patterns (and associated uncertainty) with higher spatial resolution across areas where no information is otherwise available. This approach relies on geomorphometry derived terrain parameters and machine learning models to improve the statistical accuracy and the spatial resolution (from 27km to 1km grids) of satellite soil moisture information across the conterminous United States on an annual basis (1991–2016). We derived 15 primary and secondary terrain parameters from a digital elevation model. We trained a machine learning algorithm (i.e., kernel weighted nearest neighbors) for each year. Terrain parameters were used as predictors and annual satellite soil moisture estimates were used to train the models. The explained variance for all models-years was >70% (10-fold cross-validation). The 1km soil moisture grids (compared to the original satellite soil moisture estimates) had higher correlations (improving from $r^2 = 0.1$ to $r^2 = 0.46$) and lower bias (improving from 0.062 to 0.057 m3/m3) with field soil moisture observations from the North American Soil Moisture Database (n = 668 locations with available data between 1991–2013; 0-5cm depth). We conclude that the fusion of geomorphometry methods and satellite soil moisture estimates is useful to increase the spatial resolution and accuracy of satellite-derived soil moisture. This approach can be applied to other satellite-derived soil moisture estimates and regions across the world.

## Introduction

Continuous national to continental scale soil moisture information is increasingly needed to characterize spatial and temporal trends of terrestrial productivity patterns (e.g., production of food, fiber and energy). This is because soil moisture is a key variable regulating hydrological and biogeochemical cycles, and thus studying its spatial-temporal dynamics is crucial for

**Data Availability Statement:** Data is publicly available at the ESA-CCI website and is described in the main document. We also provide an R code in: Guevara M. Vargas R. Protocol for Downscaling Satellite Soil Moisture Estimates using

Geomorphometry and Machine Learning, Protocols.io. protocols.io; 1970; https://doi:10.17504/protocols.io.6cahase, and model outputs are available at: Guevara M and Vargas R. 2019. Annual soil moisture predictions across conterminous United States using remote sensing and terrain analysis across 1 km grids (1991-2016), HydroShare, https://doi.org/10.4211/hs.b8f6eae9d89241cf8b5904033460af61.

**Funding:** MG acknowledges a fellowship from CONACyT (382790). RV acknowledges support from the National Science Foundation CIF21 DIBBs (Grant #1724843).

**Competing interests:** The authors have declared that no competing interests exist.

assessing the potential impact of climate change on water resources [1–4]. Currently, the most feasible way to obtain national to continental soil moisture information is using remote sensing. Microwave remote sensing devices deployed on multiple earth observation satellites are able to quantify the dielectric constant of soil surface and retrieve soil moisture estimates [5]. However, there are spatial gaps of satellite-based soil moisture information and its current spatial resolution (> 1km grids) limits its applicability at the ecosystem-to-landscape scales to address the ecological implications of soil moisture dynamics [5–8].

Satellite soil moisture records are an effective indicator for monitoring global soil conditions and forecasting climate impacts on terrestrial ecosystems, because soil moisture estimates are required for assessing feedbacks between water and biogeochemical cycles [9–12]. In addition, accurate soil moisture information is critical to predict terrestrial and atmospheric interactions such as water evapotranspiration or $CO_2$ emissions from soils [3, 13–15]. However, soil moisture information at spatial resolution of 1x1km pixels or less is not yet available across large areas of the world and the coarse pixel size (>1km pixels) of available satellite soil moisture records is limited for spatial analysis (i.e., hydrological, ecological) at small regional levels (e.g., county- to state). In addition, satellite soil moisture estimates are representative only of the first few 0–5 to 10 cm of top-soil surface [16]. Therefore, comparing multiple sources for satellite soil moisture and field soil moisture estimates is constantly required for precise interpretations of soil moisture spatial patterns [17–19].

There is an opportunity for exploring statistical relationships across different sources of remote sensing information (e.g., topography and soil moisture) and developing alternative soil moisture spatial datasets (i.e., grids) to improve the continental-to-global spatial resolution of current satellite soil moisture estimates [7]. Spatially explicit soil moisture estimates can be obtained across large areas with a coarse spatial resolution (between 25–50 km grids) from radar-based microwave platforms deployed across different satellite soil moisture missions [20–21]. The availability of historical soil moisture records of these sources has increased during the last decade with unprecedented levels of temporal resolution (i.e., daily from years 1978-present) at the global scale. However, large areas constantly covered by snow, extremely dry regions or tropical rain forests (where there is a higher content of water above ground) lack of precise soil moisture satellite records due to sensor intrinsic limitations (e.g., saturation or noise) across these environmental conditions [22].

One valuable product that is affected by the aforementioned environmental conditions is the ESA-CCI (European Space Agency Climate Change Initiative) soil moisture product [20–21]. The ESA-CCI mission makes rapidly available long-term soil moisture estimates with daily temporal resolution from the 1978s to date, and it represents the state-of-the-art knowledge tool for assessing long term trends in the climate system. Modeling, validation and calibration frameworks are required for improving the spatial representation of this important dataset, and for predicting soil moisture patterns across areas where no satellite estimates are available.

Currently, there is an increasing availability of fine-gridded information sources and modeling approaches that could be used for increasing the spatial resolution (hereinafter downscaling) of the ESA-CCI satellite soil moisture estimates (e.g., soil moisture predictions across <1x1km grids). Downscaling (and subsequently gap-filling) satellite soil moisture estimates has been the objective of empirical modeling approaches based on sub-grids of soil moisture related information such as soil texture [23]. Other approaches followed environmental correlation methods and generated soil moisture predictions for satellite soil moisture estimates using both data-driven or hypothesis driven models and multiple sub-grids of ancillary information [24–26]. These sub-grids of information usually include vegetation related optical remote sensing imagery, gridded soil information, land cover classes and landforms

[27–30]. Most of these approaches have been tested for specific study sites. Other studies have focused on applying a digital soil mapping approach (a reference framework for understanding the spatial distribution of soil variability [31]) and multiple upscaling methods for predicting soil moisture patterns at the continental scale [26, 32]. An overview of multiple approaches for downscaling satellite soil moisture (e.g., empirically based, physically based) has been previously discussed [33]. Here, we propose that digital terrain analysis (i.e., geomorphometry) can also be applied for empirically downscaling soil moisture satellite-based information across continental-to-global spatial scales.

Geomorphometry is an emergent discipline in earth sciences dedicated to the quantitative analysis of land surface characteristics and topography [34–35]. For geomorphometry, the analysis of topography includes the generation of a diversity of hydrologically meaningful terrain parameters (i.e., slope, aspect, curvature, valley depth index, topographic wetness index) that aim to represent how the landscape physically constrains water inputs (e.g., rainwater, irrigation, overland flow) that reaches the soil surface [35–36]. These terrain parameters are referred as "digital" because they are usually derived from digital elevation models using Geographic Information Systems (GIS). These digital terrain parameters are hydrologically meaningful because at the landscape scale, soil moisture is partially controlled by topography-related factors (i.e., slope, aspect, curvature) that physically constrain soil water inputs and soil hydraulic properties (e.g., soil texture, structure). Based on these geomorphometry principles [35–39], we propose that it is possible to determine which terrain parameters are the strongest predictors of the spatial variability of satellite soil moisture. Statistically coupling the spatial variability of satellite soil moisture with hydrologically meaningful terrain parameters could be an alternative way to improve the spatial resolution and accuracy of satellite soil moisture estimates across scales. This is possible because topography (represented by digital terrain parameters) directly affects: 1) the angle of the satellite microwave signal at the soil surface; and 2) the overall distribution of water in the landscape.

Topography is a major driver for soil moisture and topography surrogates (e.g., land form or elevation map) have been combined with other variables (e.g., climate, soils, vegetation and land use) for downscaling satellite soil moisture estimates [33]. Furthermore, previous studies have shown that realistic soil moisture patterns can be obtained using topographic information across site specific and catchment scales including physically based and empirical approaches [40–41], However, the exclusive use of geomorphometryderived products (i.e., digital terrain parameters) for downscaling satellite soil moisture estimates has not yet been explored in detail from national-to-continental scales. This approach is relevant to avoid statistical redundancies and potential spurious correlations when downscaled soil moisture is further used or analyzed with vegetation- or climate-related variables (when these aforementioned variables were used for downscaling of satellite derived soil moisture). In this study, we show the potential of a soil moisture prediction framework purely based on digital terrain parameters.

Our main objective is to generate a soil moisture prediction framework by coupling satellite soil moisture estimates with digital terrain parameters as prediction factors. Coupling the complexity of topographic gradients and the multi-temporal nature of satellite soil moisture requires an approach that should account for non-linear relationships. Machine learning approaches could account for non-linearity based on probability and the ability of computer systems to reproduce and 'learn' (i.e., decide the best solution after multiple model realizations) from multiple modeling outputs (i.e., varying model parameters of combinations of training and testing random samples) [42]. Furthermore, machine learning is now a common component of geoscientific research leading the discovery of new knowledge in the earth system [43] including mapping of soil organic carbon [44], soil greenhouse gas fluxes [8, 45] and soil moisture estimates [46].

We postulate that the data fusion between satellite soil moisture with hydrologically meaningful terrain parameters can enhance the spatial resolution, representativeness and quality (i.e., accuracy) of current coarse satellite soil moisture grids. We focus on the conterminous United States (CONUS) given the large availability of soil moisture records for validation purposes from the North American Soil Moisture Database (NASMD) [47]. The novelty of this research relies on proposing an alternative approach for obtaining soil moisture gridded estimates with no gaps and at high spatial resolution (i.e., 1km) determined by topographic prediction factors. This study is based on public sources of satellite information (derived from ESA-CCI soil moisture product) and a data-driven framework that could be reproduced and applied across the world.

## Materials and methods

Our downscaling approach relied on a Digital Elevation Model (DEM), and satellite soil moisture records. Soil moisture information was acquired from the ESA-CCI [20–21]. The development and reliability (i.e., validation) of this remote sensing soil moisture product has been documented by previous studies [20–21, 48]. Our framework includes prediction factors for soil moisture from digital terrain analysis. These terrain predictors were derived across CONUS using 1km grids. Machine learning was used for generating soil moisture predictions (annual, 1991–2016) and the satellite soil moisture estimates provided by the ESA-CCI were used as training data. Field soil moisture observations from the North American Soil Moisture dataset were used for validating the soil moisture predictions based on digital terrain analysis (Fig 1).

### Datasets and data preparation

The downscaled dataset were obtained from the ESA-CCI satellite soil moisture estimates between 1991 and 2016, and the validation dataset were field soil moisture measurements from the NASMD (S1 Fig). The downscaling framework is explained in the following sections. The ESA-CCI soil moisture product has a daily temporal coverage from 1978 to 2016 and a spatial resolution of ~27 km (S2 Fig). Among several remotely sensed soil moisture products [16, 49–53], we decided to use the ESA-CCI soil moisture product because it covers a larger period of time compared with other satellite soil moisture products (e.g., NASA SMAP). We highlight that satellite soil moisture information is used for training a machine learning model for each year, and independent field soil moisture records area only used for validating the downscaled soil moisture predictions.

For externally validating, we used the NASMD because it has been curated following a strict quality control calibrated for CONUS [44] (S1 Fig). This data collection effort consists of a harmonized and quality-controlled soil moisture dataset with contributions from over 2000 meteorological stations across CONUS described in previous studies [47]. The NASMD also includes records of soil moisture registered in the International Soil Moisture Network (ISMN) [47, 54]. The NASMD provides processed data from each station location in each network following a standardization framework focused in North America [47–19]. We used soil moisture records at 5 cm of depth (n = 5541 daily measurements) from 668 stations with available (from the aforementioned sources) soil moisture estimates at this depth because radar-based soil moisture estimates are representative for these first few centimeters of topsoil surface [16].

As prediction factors for soil moisture, we calculated hydrologically meaningful terrain parameters for CONUS (S1 Table) using information from a radar-based DEM [55–56]. These terrain parameters are quantitative spatial grids representing the topographic variability that

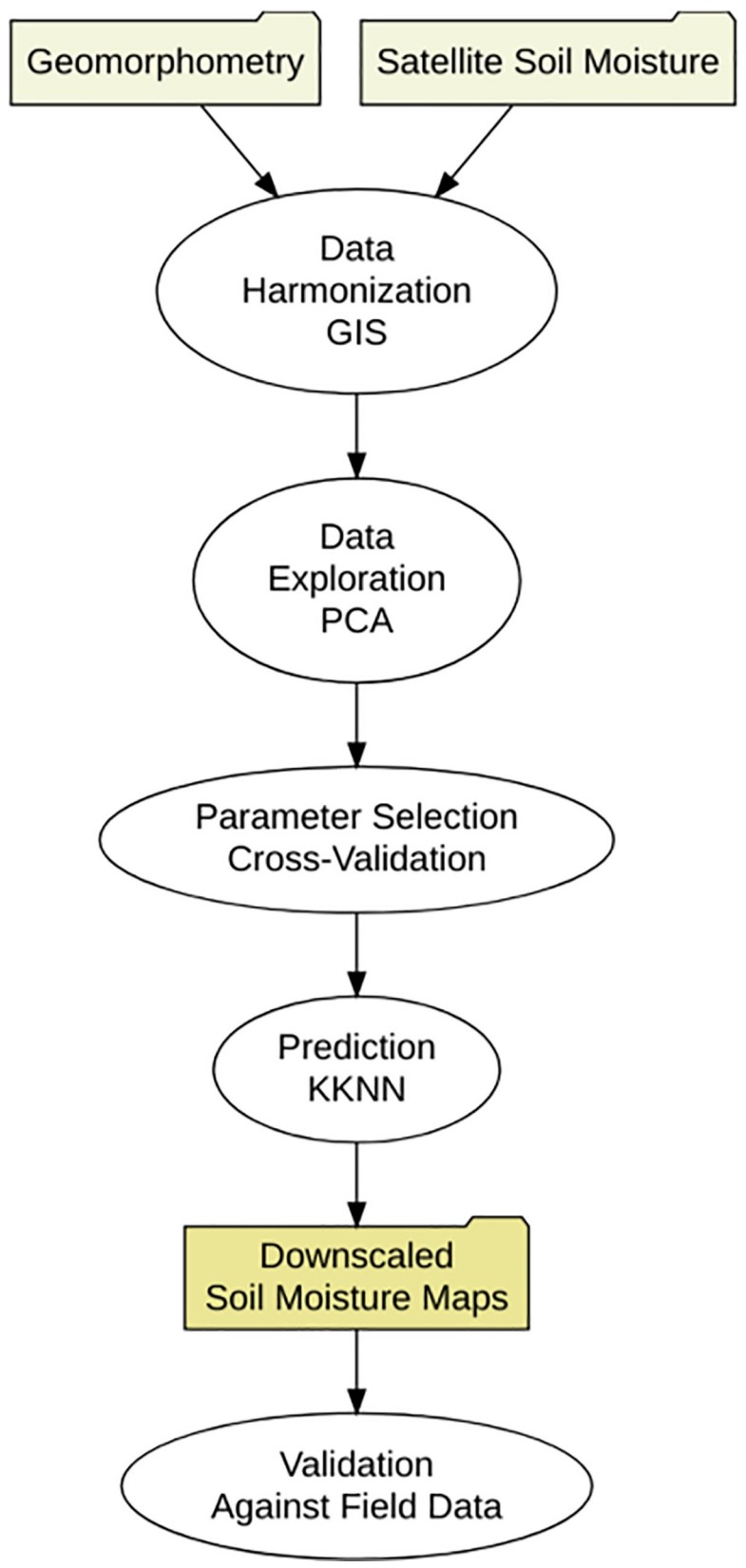

**Fig 1. Soil moisture prediction framework.** The folders are the inputs and outputs and the ovals are methods for data preparation (data bases harmonization), modeling (for prediction) and validation (for assessing the reliability of soil moisture maps). The field data from the North American Soil Moisture Database (NASMD) was only used for validation purposes (i.e., not for training the model).

directly influence the water distribution across the landscape [35], which supports the physical link between soil moisture and topography. These parameters were the basis for downscaling satellite soil moisture records to 1km grids. This spatial resolution captures the major variability of topographic features across CONUS and is commonly used on large-scale ecosystem studies and soil mapping efforts [56–57].

For the calculation of soil moisture prediction factors, we used automated digital terrain analysis using the System for Automated Geographical Analysis-Geographical Information System (SAGA-GIS) [36]. The automated implementation of SAGA-GIS for Geomorphometry (module for basic terrain analysis) includes a preprocessing stage to remove spurious sinks and reduce the presence of other artifacts in the elevation gridded surface (e.g., false pikes or flat areas). After preprocessing the DEM, fifteen hydrologically meaningful terrain parameters were generated for the CONUS from elevation data including primary (i.e., slope, aspect) and secondary parameters (i.e., cross-sectional curvature, longitudinal curvature, analytical hillshading, convergence index, closed depressions, catchment area, topographic wetness index, length-slope factor, channel network base level, vertical distance to channel network, and valley depth index; Fig 2).

Primary terrain parameters are direct descriptions of elevation data, for example slope and aspect, which are respectively the first and second derivative of elevation data. Secondary parameters are generated by combining a primary terrain parameter with a mathematical formulation to describe process controlled by topography that are directly related to soil moisture variability [34–35, 37], such as the overland flow accumulation. The topographic wetness index for example, indicates areas where water tend to accumulate by effect of topography and is a secondary index derived by the combination upslope area draining through a certain point per unit contour length and the local terrain slope [34–35]. The values of these terrain parameters (S1 Table) were extracted for each one of the ESA-CCI soil moisture pixels using as reference the central coordinates of each ESA-CCI pixel (Fig 1 inputs).

## Data exploration

We used a principal component analysis (PCA) prior to modeling for data exploration and description of general relationships between soil moisture values and topography (represented by the aforementioned terrain parameters). The purpose was to simplify the dimensionality of the data set to identify the main relationships (between soil moisture and topographic parameters) driving our downscaling framework (Fig 1 methods). The PCA was implemented as in previous work [58], based on a reference value representing the 0.95-quantile of the variability obtained by randomly simulating 300 data tables of equivalent size on the basis of a normal distribution. This analysis was applied to the terrain parameters at the locations of the field stations in order to compare the relationship of the first PCA and the values of soil moisture from the ESA-CCI grids and from the field data.

## Model building

For this study we built a model for each annual mean of satellite soil moisture grids. We used a machine learning kernel-based model (kernel weighted nearest neighbors, kknn) [59–60] to generate predictions and downscale satellite soil moisture grids (Fig 1 methods). The training

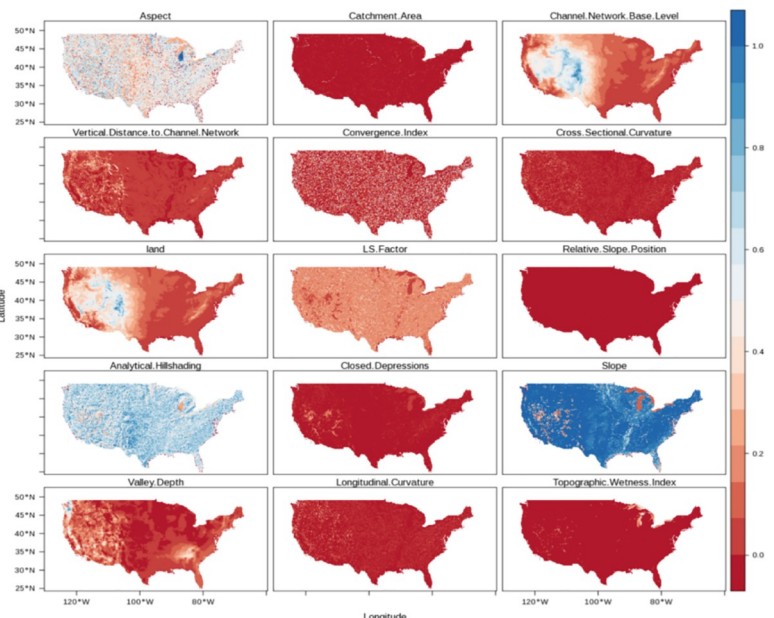

**Fig 2. Elevation and hydrologically meaningful terrain parameters at 1x1km of spatial resolution derived using the standard SAGA-GIS basic terrain parameters module.** These maps were normalized (between 0–1) and then used as prediction factors to downscale soil moisture across CONUS.

dataset for each model/year were the annual mean values of the ESA-CCI soil moisture product. The kknn model has two main model parameters: the optimum number of neighbors (k) and the optimal kernel function (okf). First, we defined k, which is the number of neighbors to be considered for the prediction. Second, we selected the okf, which is a reference (e.g., triangular, epanechnikov, Gaussian, optimal) for the probability density function of the variable to be predicted. The okf is used to convert distances (i.e., Minkowski distance) into weights used to calculate the k-weighted average. These kknn model parameters (k and okf) were selected by the means of 10-fold cross validation as previously recommended [61]. Cross-validation is a well-known re-sampling technique that divides data into 10 roughly equal subsets. For every possible parameter value (e.g., *k* from 1 to 50 and okf [triangular, epanechnikov, Gaussian, optimal]), 10 different models are generated, each using 90% of the data then being evaluated on the remaining 10%. To predict soil moisture information at 1 km of spatial resolution for each year (between 1991 and 2016), we selected the combination of optimal k and okf that lead to the highest correlation (between observed and predicted data) with the lowest root mean squared error (RMSE) after the cross-validation strategy. Thus, for each year we were able to predict soil moisture across 1x1 km grids (Fig 1 *outputs*).

## Validation using field observations across CONUS

Downscaled soil moisture grids were compared against field measurements and we computed the explained variance ($r^2$) using a linear fit (observed *vs* predicted) for each field soil moisture location. Given the relatively low density and sparse spatial distribution of field data for validating (S1 Fig), we bootstrapped the independent validation using different sample sizes (from 10 to 100% of data with increments each 10%) to avoid systematic bias associated with the spatial distribution and density of field soil moisture information. We sampled (n = 1000) repeatedly the original and the downscaled soil moisture grids aiming to identify their correlation with the aforementioned validation dataset (i.e., observed vs predicted).

We also computed the spatial structure (spatial autocorrelation) of the explained variance (correlation between geographical distance and variance of $r^2$ values) for estimating an $r^2$ map using an interpolation technique known in geostatistics as Ordinary Kriging [62]. Ordinary Kriging is a well-known method for spatial interpolation based on the spatial structure or spatial autocorrelation of the variable of interest (the $r^2$ values between the field observations and the predicted soil moisture values). The spatial autocorrelation is defined by the relationship between geographical distances and variance of values at a given distance, and it is commonly characterized using variograms. We followed an automated variogram parameterization (the optimal selection for the variogram parameters nugget, sill and range required to perform Ordinary Kriging) proposed in previous work [63].

As implemented in the automap package of R [63], the initial sill is estimated as the mean of the maximum and the median values of the semi-variance. The semi-variance is defined by the variance within multiple distance intervals. For modeling the spatial autocorrelation this algorithm iterates over multiple variogram model parameters selecting the model (e.g., spherical, exponential, Gaussian) that has the smallest residual sum of squares with the sample variogram. The initial range is defined as 0.10 times the diagonal of the bounding box of the data. The initial nugget is defined as the minimum value of the semi-variance. Thus, the parameters used for obtaining a continuous map showing spatial trends in the $r^2$ were: a Gaussian (normal) model form, a nugget value of 0.06 m$^3$ m$^{-3}$, a sill of 0.08 m$^3$ m$^{-3}$ and an approximate range of 428.7 km. This map was generated because it could provide insights about overall sources of modeling errors (e.g., environmental similarities in multiple areas showing low or high explained variance) and their spatial distribution. All analyzes were performed in R [64] using public sources of data. Our protocol and R code used for generating the soil moisture predictions at 1km grids is available online (http://dx.doi.org/10.17504/protocols.io.6cahase) for reproducibility of this research [65].

## Results

The exploratory PCA showed that the first two PCs explained 33% of the total dataset variability (S3A Fig), where the first PC explained 18% of total variability and at least five PCs were needed to explain 70% of total variability. The first PC was best correlated with elevation (r = 0.82) and with the vertical distance to channel network (r = 0.88). Elevation varied negatively with soil moisture, as well as other secondary terrain parameters such as the base level channel network elevation (distance from each pixel to the closer highest point), while the valley depth index varied positively with soil moisture (S3B Fig). The relative slope position (indicating the dominance of flat or complex terrain) and the topographic wetness index (which indicates areas where water tends to accumulate) were also correlated with soil moisture across the first 5 PCs. Thus, multiple terrain parameters varied positively and negatively with soil moisture values (S1 Appendix).

Our framework to predict soil moisture based on topography and remote sensing was able to explain, on average 79±0.1% of the variability of satellite soil moisture information as revealed by the cross-validation strategy. The root mean squared error (RMSE) derived from the cross-validation varied around 0.03 m$^3$/m$^3$, while the percentage of explained variance was in all cases above 70% (Table 1).

By applying the model coefficients to the topographic prediction factors across CONUS, we generated 26 cross-validated maps (for years 1991–2016) of mean annual soil moisture estimates within 1x1km grids (Fig 3). The downscaled product shows a higher level of spatial variability due the increased spatial detail achieved by downscaling soil moisture to 1x1km grids (S4 Fig). Our predictions reveal a clear bimodal distribution of soil moisture values (e.g., from the east to the west, Fig 4) which is also evident in the original estimate (S5 Fig). The statistical

**Table 1. The cross-validation results for each year.** This table shows the correlation, root mean squared error (RMSE) in m$^3$/m$^3$, the number of training data available (*n*), the optimal kernel function (*okf*), and the optimal number of neighbors used for predicting to new data (*k*).

| Model | Year | Correlation | RMSE | *n* | *okf* | *k* |
|-------|------|-------------|------|-----|-------|-----|
| 1 | 1991 | 0.85 | 0.03 | 18058 | triangular | 18 |
| 2 | 1992 | 0.89 | 0.03 | 18429 | triangular | 16 |
| 3 | 1993 | 0.88 | 0.03 | 18107 | triangular | 18 |
| 4 | 1994 | 0.9 | 0.03 | 18367 | triangular | 16 |
| 5 | 1995 | 0.88 | 0.03 | 18385 | triangular | 18 |
| 6 | 1996 | 0.9 | 0.03 | 18454 | triangular | 15 |
| 7 | 1997 | 0.88 | 0.03 | 18428 | triangular | 15 |
| 8 | 1998 | 0.88 | 0.03 | 18540 | triangular | 16 |
| 9 | 1999 | 0.89 | 0.03 | 18542 | triangular | 15 |
| 10 | 2000 | 0.9 | 0.03 | 18547 | triangular | 15 |
| 11 | 2001 | 0.9 | 0.03 | 18523 | triangular | 15 |
| 12 | 2002 | 0.9 | 0.03 | 19170 | triangular | 16 |
| 13 | 2003 | 0.89 | 0.03 | 19132 | triangular | 16 |
| 14 | 2004 | 0.89 | 0.03 | 18934 | triangular | 16 |
| 15 | 2005 | 0.89 | 0.03 | 19132 | triangular | 16 |
| 16 | 2006 | 0.9 | 0.03 | 19131 | triangular | 16 |
| 17 | 2007 | 0.88 | 0.03 | 19142 | triangular | 16 |
| 18 | 2008 | 0.9 | 0.03 | 19136 | triangular | 16 |
| 19 | 2009 | 0.9 | 0.03 | 19142 | triangular | 16 |
| 20 | 2010 | 0.88 | 0.03 | 19245 | triangular | 18 |
| 21 | 2011 | 0.9 | 0.03 | 19255 | triangular | 18 |
| 22 | 2012 | 0.9 | 0.03 | 19252 | triangular | 16 |
| 23 | 2013 | 0.89 | 0.03 | 19226 | triangular | 16 |
| 24 | 2014 | 0.89 | 0.03 | 19227 | triangular | 16 |
| 25 | 2015 | 0.88 | 0.03 | 19231 | triangular | 16 |
| 26 | 2016 | 0.88 | 0.03 | 19225 | triangular | 16 |

comparison between the original product and the downscaled product shows a high level of agreement with an r$^2$ value of 0.72.

We provided a visual comparison between the original satellite estimate and the downscaled results including both median (Fig 4A and 4B) and standard deviation values (Fig 4C and 4D). We also show the uncertainty of the original soil moisture product as reported by its developers (Fig 4E) and the r$^2$ map from the validation against field stations (Fig 4F). This r$^2$ map (Fig 4F) is based on the spatial autocorrelation found in the variogram estimation (e.g., sill > nugget) applied to the r$^2$ values of the data used for validating our approach (S1 Fig). This r$^2$ map is indicating areas with similar soil moisture conditions regulating the spatial variability of soil moisture and affecting the performance of our models, around the sites of available field data for validating our approach. The r$^2$ map shows the lowest values across the Central Plains of the US and the lower Mississippi basin. The lower values in the r$^2$ map are consistent with the high uncertainty values of the original satellite estimate (Fig 4E).

The r$^2$ map in Fig 4F provided insights about the relationship between soil moisture gridded surfaces and soil moisture field data. Higher r$^2$ values were found across the east coast, the Northern Plains and water-limited environments across the western states. We found that our soil moisture downscaled output better correlates (nearly 25% improvement) with NASMD field observations when compared to the original soil moisture satellite estimates (Fig 5).

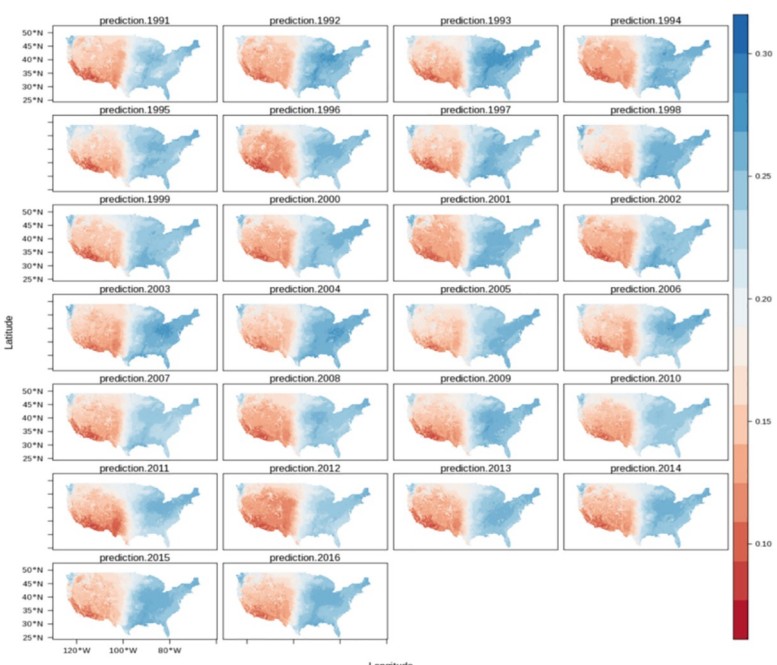

**Fig 3. Annual means of soil moisture (1991–2016) downscaled to 1x1km grids across CONUS using terrain parameters as prediction factors.**

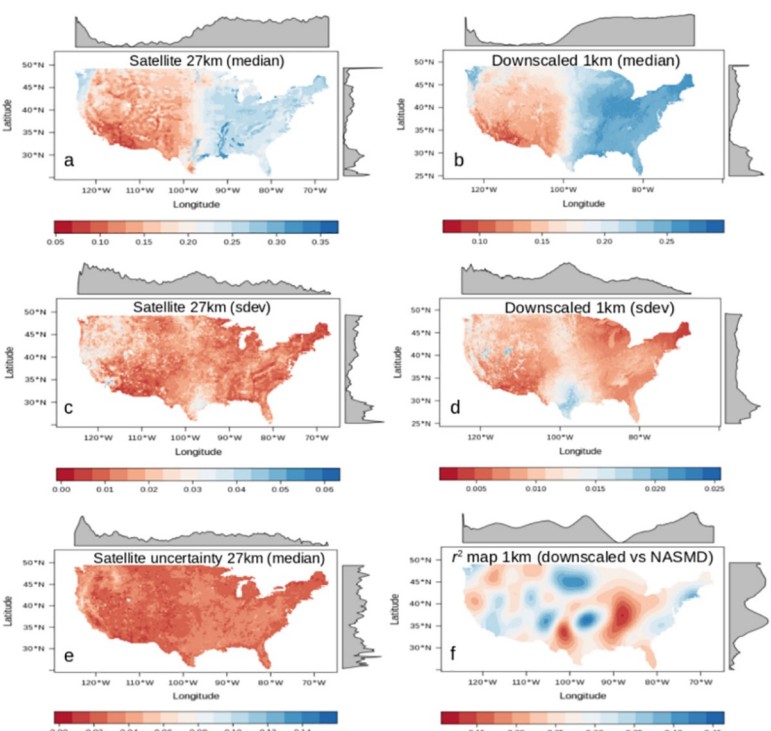

**Fig 4. Comparison of the original (27km grids) and the downscaled (1km grids) soil moisture products.** Median (a, b) and standard deviation (c, d, sdev) values of satellite soil moisture and downscaled soil moisture values (1991–2016). Uncertainty reported by the ESA-CCI soil moisture (e) and the explained variance map ($r^2$) between field data and downscaled soil moisture (f).

This improvement was consistent after repeating it using random samples and different sample sizes (from 10 to 90% of available validation data) from the NASMD field observations (Fig 5). Consistently, we found a negative mean bias (surrogate of systematic error) in the ESA-CCI when compared against field stations of the NASMD (-0.051) that is slightly lower when comparing the downscaled soil moisture predictions against the NASMD (-0.048). The resulting RMSE of validating against field data was also slightly lower (0.057 m3/m3) for the downscaled estimate compared with the original product (0.062 m3/m3). However, there is a sparse distribution of validation data and large areas of CONUS lack of field information for validating/calibrating soil moisture predictions (Fig 6). Considering the quality-controlled records available from the NASMD across CONUS and the coarse scale of the ESA-CCI soil moisture product, our approach suggests an improvement in the spatial resolution (from 27 to 1km grids) of soil moisture estimates while maintaining the integrity of the original satellite values.

The original satellite values, the downscaled product and the ISMN dataset showed a similar correlation with the terrain predictors. For example, the first PCA (represented by the distance to channel network and elevation), was negatively correlated with field soil moisture, the satellite original product and our soil moisture predictions. The correlation values were r = -0.17, r = -0.27, and r = -0.28 respectively. These relationships showed a similar pattern in the statistical space (Fig 7).

## Discussion

Our soil moisture downscaling framework was able to improve the spatial detail of ESA-CCI satellite soil moisture product and its agreement with field soil moisture records from the

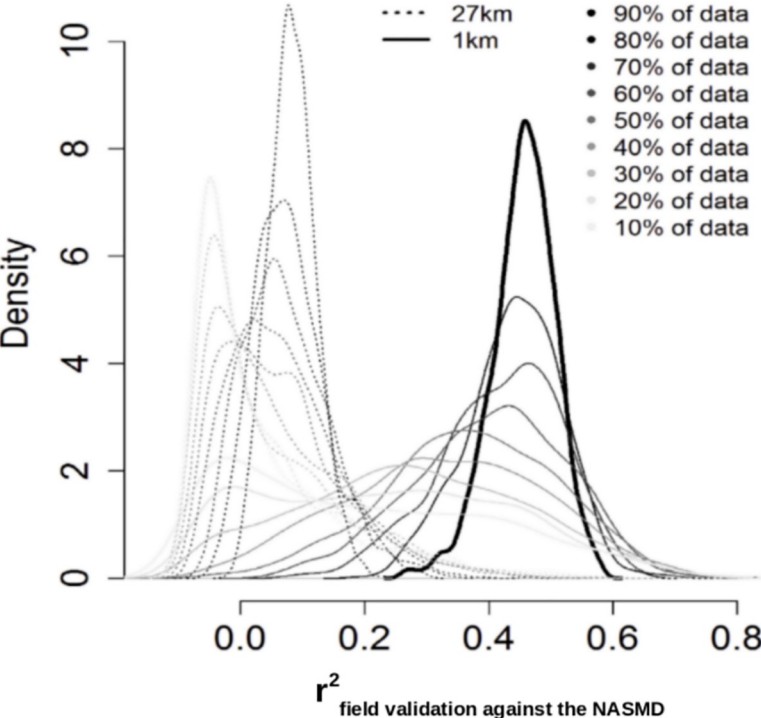

**Fig 5. Validation of soil moisture gridded estimates (original 27 and 1km grids) against NASMD field observations.** Dashed line represents the relationship of field stations and soil moisture gridded estimates at 27x27km, while black line represents the relationship between field stations and the downscaled 1x1km soil moisture product. In all cases (all sample sizes), the 1x1km product showed higher $r^2$ with the NASMD than the ESA-CCI soil moisture estimates.

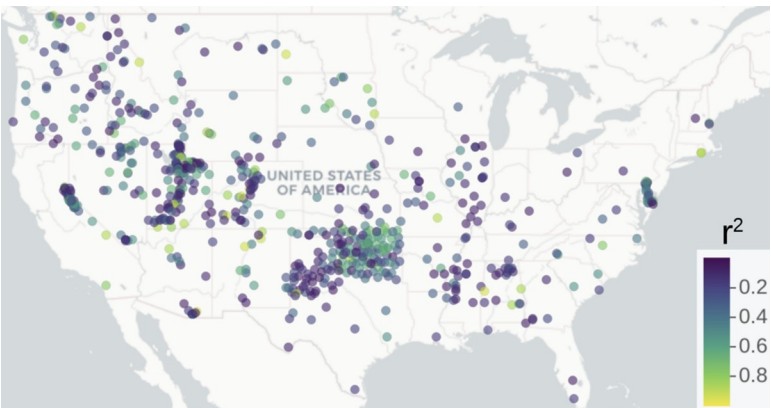

**Fig 6. Explained variances computed for each meteorological station of the NASMD and the corresponding pixel of our soil moisture predictions based on geomorphometry.**

NASMD. It is well known that topography has a direct influence on the overall water distribution across the landscape [38–39] and in the angle between satellite retrieval and the Earth's surface. Thus, we demonstrated how a coarse scale satellite-based soil moisture product (27x27km of spatial resolution), in combination with hydrologically meaningful terrain parameters, can be coupled using machine learning algorithms to generate a fine-gridded and gap-free soil moisture product at the annual scale across CONUS.

We found a correlation between field soil moisture estimates and topography that is similar to the correlation between satellite estimates and topography (Fig 7), suggesting that topography can be an effective predictor for direct soil moisture measurements (i.e., from microwave remote sensing). Previous studies have confirmed this correlation between topographic variability and soil moisture conditions for downscaling soil moisture across multiple catchment scales and environmental conditions [40–41]. We recognize that our modeling approach based on digital terrain analysis does not directly account for local variations of evaporation, soil structure or vegetation (but indirectly). Our approach assumed that topography (described by the shape of multiple digital terrain parameters) is also capturing some variability associated to all factors affecting soil moisture. In contrast to previous downscaling efforts using vegetation and climate information [33, 66], we generated 26 annual soil moisture predictions (1991–2016, 1x1 km of spatial resolution) that are independent of ecological data (i.e., vegetation greenness) and climate information, (i.e., precipitation and temperature). This topography-based approach is able to maintain the original satellite soil moisture statistical distribution (S5 Fig) and has the advantage that our modelling output could be further related to independent datasets of ecological or climate variables [67–68] and avoid subsequent spurious relationships.

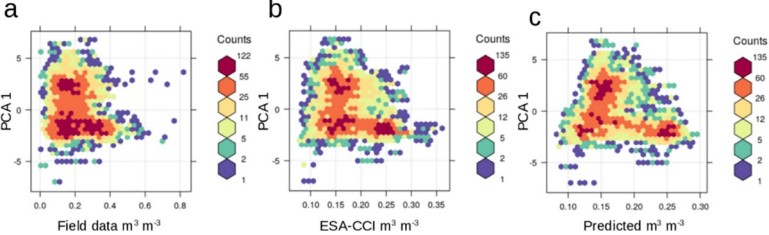

**Fig 7.** Relationships between the first PC of terrain parameters with soil moisture field data (a), with the ESA-CCI satellite product (b), and with the soil moisture predictions based on terrain parameters (c).

We showed that this topography-based approach is able to improve the correlation of the original estimate when compared against field data from the NASMD while maintaining the integrity of its values (i.e., same systematic error between the NASMD and the original or the downscaled product). Therefore, we provided a reliable (topography-based) approach to predict the satellite soil moisture patterns across finer spatial grids and in areas where no satellite soil moisture is available.

The downscaling process of satellite soil moisture from 27 to 1km grids across CONUS is supported on both internal (Table 1) and independent (Fig 5) validation frameworks to describe modeling performance. The accuracy of our modeling framework showed explained variances >70% and RMSE values considerably below (~0.03 $m^3$ $m^{-3}$) the satellite soil moisture mean of 0.22 $m^3$ $m^{-3}$, which is suitable for many applications [66], such as the detection of irrigation signals [69]. Similar results have been found recently for specific study sites [70]. Our results obtained by the cross-validation strategy and ground validation supports the application of a topography-based model to predict satellite soil moisture estimates (Fig 4).

Our results showed that higher soil moisture values could be found across lower elevations, areas with generally large and gentle slopes mainly across valley bottoms and across catchment areas where water tends to accumulate. This interpretation could explain the short distance in the multivariate analysis of satellite soil moisture estimates to elevation and derived terrain parameters such as the vertical distance (of each pixel) to the nearest channel network, the valley depth index and the topographic wetness index. The multivariate analysis also suggested some degree of statistical redundancy between the topographic prediction factors (S1 Appendix) as they were derived from the digital elevation model by the means of geomorphometry [34–39]. For example, we found that the topographic wetness index is highly correlated with the length-slope factor (>0.80%), and this is because they are two secondary parameters that depend on slope (primary terrain attribute) [35]. Elevation and slope are respectively required for calculating secondary terrain parameters such as the valley depth index and the topographic wetness index [36] and these terrain parameters varied closely with soil moisture in the multivariate space (S1 Appendix). Thus, understanding the main relationships between topographic prediction factors and soil moisture can be useful for reducing modeling complexity while increasing our capacity to interpret modeling results.

The spatial detail of soil moisture estimates using 1km grids across the continental scale of CONUS is consistent with the variability of soil moisture patterns between the western and eastern United States. While drought scenarios have been recently reported for the western states [71] evidence of precipitation increase has been reported recently in the eastern states [72]. Our soil moisture downscaled estimates (Fig 3) revealed a soil moisture gradient across the Central Plains of CONUS and a clear separation of two major soil moisture data populations (i.e., soil moisture values with a bimodality distribution) from the drier west, to the humid east (S5 Fig).

The original satellite soil moisture estimates also show this bimodal distribution but with a lesser extent (S2 Fig). The bimodal distribution of soil moisture could be explained by a negative soil moisture and precipitation feedback in the western CONUS and a positive soil moisture and precipitation feedback in the eastern CONUS [68]. Furthermore, areas with soil moisture bimodality have been recognized across global satellite observations and climate models [73]. We identified areas of low agreement between our soil moisture predictions and field stations (lower $r^2$ values) across the transitional ecosystems (Fig 4) from drier to humid soil moisture environments (i.e., Central Plains and lower Mississippi basin). It is likely that these transitional areas drive changes in water availability in surface and subsurface hydrological systems [74]. The lower Mississippi basin, specifically the area across the surroundings of the Mississippi delta, is an example of a transitional area experiencing aquifer depletion [75]

where both flooding events and droughts tend to occur within shorter distances that are not captured by the original satellite soil moisture information. These are the type areas where we found lower values of agreement ($r^2$ values) between satellite and ground soil moisture observations. These low correlation values can be also explained by the use of multiple soil moisture networks with different types of sensors and measurement techniques [19]. Also, the imperfections of prediction factors used for soil moisture spatial variability models represent a potential source of uncertainty.

As any downscaling effort dependent on covariates (i.e., terrain parameters), our approach is vulnerable to data quality limitations such as the presence of systematic errors on these covariates. Other errors are derived from input data imperfections and difficulties meeting modeling assumptions. These errors in soil moisture modeling inputs increase the risk of bias and uncertainty propagation to subsequent soil moisture modeling outputs and soil mapping applications [76–78]. For example, elevation data surfaces derived from remote sensing data (such as the global DEM used here) could show artifacts (i.e., false pikes or spurious sinks) due to data saturation or signal noise that can be propagated to final soil moisture predictions [79]. We minimized this issue by using SAGA-GIS [36] as it has adopted methods for preprocessing and perform DEM quality checks [80] before deriving the topographic prediction factors used in this study. Because input covariates could not be fully free of errors, we advocate for reporting information on bias and $r^2$ values to inform about accuracy (e.g., Table 1) as important components for interpreting soil moisture predictions.

Our results suggest that the original coarse scale soil moisture product and the values of soil moisture from the NASMD (Fig 5) are difficult to compare in terms of spatial variability, as is highlighted in previous studies [19]. This is because a satellite soil moisture pixel from the ESA-CCI product provides a value across a larger area (27x27km) than a field measurement at a specific sampling location (defined by geographical coordinates). This scale dependent effect (27x27km vs 1:1 field scale) is reduced (>25%) with soil moisture predictions across finer grids (1km). The downscaled soil moisture maps showed a higher agreement with field soil moisture records from the NASMD (Fig 5), supporting the applicability of this soil moisture product for applications that required higher spatial resolution.

Our soil moisture predictions across 1km grids suggest that topography can be effectively used to improve the spatial detail and accuracy of satellite soil moisture estimates. Several studies have highlighted differences in spatial representativeness between ground-based observations and satellite soil moisture products [77, 81]. Other studies have shown that the spatial representativeness of the ESA-CCI soil moisture compared with field observations is higher from regional-to-continental scales than from ecosystem-to-landscape scales [82–83]. Therefore, large uncertainties of soil moisture spatial patterns (below 1km grids) needs to be resolved for assessing and better understanding the local variability of soil moisture trends. We argue that currently there is an increasing availability of high-quality digital elevation data sources with high levels of spatial resolution (e.g., 1–2 to 30 to 90m grids) across large areas of the world [84–85] that can be used to derive reliable hydrologically meaningful terrain parameters for predicting soil moisture. The relationship of these digital terrain parameters and field soil moisture (i.e., meteorological stations) is similar to the relationship between terrain parameters and satellite soil moisture gridded estimates (Fig 7).

From a single information source (a remotely sensed DEM), we downscaled satellite records of soil moisture using a framework that theoretically is reproducible across multiple scales. The ultimate goal of reducing the multiple information sources for predicting soil moisture is to reduce the statistical redundancy in further modeling efforts (i.e., land carbon uptake models) and large-scale ecosystem studies (i.e., ecological niche modeling) that combine similar prediction factors for soil moisture (i.e., climate or vegetation indexes). These include

models estimating water evapotranspiration trends [86] and process based global carbon models that could also benefit from more accurate and independent soil moisture inputs [78]. To improve the spatial representativeness of satellite soil moisture estimates, the number of studies developing new downscaling approaches based on prediction factors is rapidly expanding [26, 28, 66, 87]. There is a pressing need to solve the current uncertainty of soil moisture estimates to accurately understand how soil moisture is limiting the primary productivity of terrestrial ecosystems [6]. Previous studies have identified a topographic signal in satellite soil moisture [88] supporting the reliability of our modeling approach. Therefore, our results provide an alternative applicable to continental scales for downscaling satellite soil moisture estimates based on hydrologically meaningful terrain parameters.

The novelty of this approach is that it could be applicable to multiple temporal resolutions (e.g., monthly or daily) as it generates independent models for each period of interest and at multiple spatial scales as the availability of terrain parameters for modeling purposes has increased substantially (i.e., meters) in the last decade. Increasing the temporal resolution of downscaled maps (i.e., from annual to monthly predictions) is beyond the scope of this study, will increase computational costs, but are theoretically possible following this approach. While monthly or weekly (or even daily soil moisture datasets) are valuable sources for large scale earth system modeling, annual averages are also valuable for detecting long term trends in the climate-land system. Rather than focusing on temporal variability of soil moisture, our results provide insights for improving the spatial variability and consequently the spatial representation of soil moisture gridded surfaces derived from satellite information.

## Conclusion

Recent studies highlight the necessity of detailed soil moisture products to account for soil moisture limitation in terrestrial ecosystems. We developed a geomorphometry-based framework to couple satellite soil moisture records with hydrologically meaningful terrain parameters. This approach is useful to avoid statistical redundancies when downscaled soil moisture is further used or analyzed with vegetation- or climate-related variables not included in the downscaling framework. We predicted (i.e., downscaled) soil moisture using 1x1km grids across CONUS at an annual scale from 1991 to 2016. This gap-free soil moisture product improved the spatial detail of the original satellite soil moisture grids and the overall agreement (increased by >20%) of these grids with the NASMD field soil moisture records. Our findings suggest that digital terrain analysis can be applied to elevation data sources to derive hydrologically meaningful terrain parameters and use these parameters predict soil moisture spatial patterns. Our framework is reproducible across the world because it is based on publicly available DEMs, ground and satellite soil moisture data. Our protocol and R code is available online (http://dx.doi.org/10.17504/protocols.io.6cahase) as well as input parameters and annual means of soil moisture at 1km grids (https://www.hydroshare.org/resource/b8f6eae9d89241cf8b5904033460af61/).

## Supporting information

**S1 Appendix. This file (AppendixS1.html) contains the results of the automated PCA report described in the methods section, for exploring relationships between soil moisture and topography.**
(HTML)

**S1 Fig. Spatial distribution of the NASMD validation set with information for the first 0-5cm depth (n 668).**
(TIF)

**S2 Fig. ESA-CCI satellite soil moisture across CONUS (~27km grids).**
(TIF)

**S3 Fig. PCA map of the first and second principal components.** The individual point cloud across the plane between the first and second PCA (a). The orthogonal relationship of the variables with higher contribution to this plane (b). Soil moisture is represented by the dotted blue line. An interpretation of these can be found in S1 Appendix.
(TIF)

**S4 Fig. Mean soil moisture for the year 2016.** Comparison between the original satellite soil moisture (a) and soil moisture predicted at 1km grids (b).
(TIF)

**S5 Fig.** Bimodal distribution of satellite soil moisture (black) and the downscaled soil moisture estimates (red).
(TIF)

**S1 Table. This table (S1 Table) contains a description of primary and secondary terrain parameters used in this study for generating soil moisture predictions.**
(DOCX)

## Author Contributions

**Conceptualization:** Mario Guevara, Rodrigo Vargas.

**Data curation:** Mario Guevara.

**Formal analysis:** Mario Guevara.

**Supervision:** Rodrigo Vargas.

**Writing – original draft:** Mario Guevara.

**Writing – review & editing:** Rodrigo Vargas.

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
