## [Decision Letter · Decision Letter 0]

2 Aug 2019

PONE-D-19-17266

Downscaling Satellite Soil Moisture using Geomorphometry and Machine Learning

PLOS ONE

Dear Dr. Vargas,

Thank you for submitting your manuscript to PLOS ONE. After careful consideration, we feel that it has merit but does not fully meet PLOS ONE’s publication criteria as it currently stands. Therefore, we invite you to submit a revised version of the manuscript that addresses the points raised during the review process.

We would appreciate receiving your revised manuscript by Sep 16 2019 11:59PM. To enhance the reproducibility of your results, we recommend that if applicable you deposit your laboratory protocols in protocols.io, where a protocol can be assigned its own identifier (DOI) such that it can be cited independently in the future. For instructions see: http://journals.plos.org/plosone/s/submission-guidelines#loc-laboratory-protocols

We look forward to receiving your revised manuscript.

Kind regards,

Benjamin Poulter

Academic Editor

PLOS ONE

Journal Requirements:

"MG acknowledges a fellowship from CONACyT. RV acknowledges support from the National Science Foundation CIF21 DIBBs (Grant #1724843)."

"RV acknowledges support from the National Science Foundation CIF21 DIBBs (Grant #1724843). "

Please provide an amended Funding Statement that declares *all* the funding or sources of support received during this specific study (whether external or internal to your organization) as detailed online in our guide for authors at http://journals.plos.org/plosone/s/submit-now Please state what role the funders took in the study.  If any authors received a salary from any of your funders, please state which authors and which funder. If the funders had no role, please state: "The funders had no role in study design, data collection and analysis, decision to publish, or preparation of the manuscript."

Additional Editor Comments:

Dear Authors,

I have received two reviews for your manuscript, both consider the work interesting and important for publication. One reviewer has several detailed questions that need to be addressed and will improve the manuscript's clarity and robustness. Consider the decision between and minor and major revision as the responses need to be addressed in detail.

Ben

Reviewers' comments:

Reviewer's Responses to Questions

**Comments to the Author**

1. Is the manuscript technically sound, and do the data support the conclusions?

Reviewer #1: Yes

Reviewer #2: Partly

2. Has the statistical analysis been performed appropriately and rigorously? 

Reviewer #1: Yes

Reviewer #2: No

3. Have the authors made all data underlying the findings in their manuscript fully available?

Reviewer #1: Yes

Reviewer #2: No

4. Is the manuscript presented in an intelligible fashion and written in standard English?

Reviewer #1: Yes

Reviewer #2: Yes

5. Review Comments to the Author

Reviewer #1: I enjoyed reading the paper by Guevara et al., which addresses the downscaling of soil moisture data using geo-morphometry. The work is useful and very well explained in the manuscript. I recommend to accept it with few minor comments:

1. Authors can add some references about different disaggregation methods applied on soil moisture datasets for downscaling, specifically related to topography or geomorphology, like:

Pellenq et al 2003, A disaggregation scheme for soil moisture based on topography and soil depth

2. Line 193: It will be nice to provide explicitly what are the primary hydrologically meaningful terrain parameters authors are referring to.

3. A little description of these terrain parameters are required to be discussed.

4. It will be great to add some discussion about the systematic error (Bias)

Reviewer #2: The manuscript by Guevara and Vargas, entitled 'Downscaling Satellite Soil Moisture using Geomorphometry and Machine Learning', covers an interesting topic appropriate for PLOS ONE. The authors developed an approach to downscale the annual soil moisture at 25 km resolution derived from ESA CCI satellite products to 1 km resolution using a machine learning approach. A set of geomorphological parameters were applied to train the model using a k-NN algorithm. I think the study is original in the sense that it provides an alternative approach for downscaling methodologies. However, I feel the downscaled results were not well justified and the writing are not clear and can be strengthened. I have several concerns as listed below:

- The title is a bit misleading as it's too generalized. The current title doesn't suggest for what temporal resolution (annual or monthly), nor for what regions this approach can be applied. Validating the results for the US continent doesn't necessarily prove that it can be applied in other regions that have different climate (e.g. high latitudes and tropics).

- The calculation of terrain parameters. It is surprised to see that some of the parameters in Figure 2 show fairly low values with almost none spatial heterogeneity. Is that a rendering problem or due to the normalization? This seems incorrect. For example, the wetness index is supposed to have significant spatial heterogeneity. See HYDRO1K or HydroSHEDS products for some of these parameters.

- The downscaled 1k soil moisture. The results don’t suggest that it maintain the original values of the satellite product. Why the downscaled results seem to have a systematical lower value than the original one? Is that because of the calculation of terrain parameters? Also, the spatial heterogeneity is not higher in the downscaled product as it’s supposed to be. Downscaled products are supposed to provide more details at a higher resolution. The legend of Figure 3 and Figure S2 are not consistent, which makes it hard to compare the downscaled with the original visually.

- The parameters used in the machine learning. This study uses a few terrain parameters to train their model - Here is an implicit assumption that the soil moisture is mainly controlled by geomorphological information. But soil moisture is proved to be strongly influenced by vegetation via its regulation of evapotranspiration.

Specific comments:

Paragraph Line58-68. It seems the first outline sentence doesn’t match the following content.

Line 164-165: need to be more specific on how you did the downscale.

Line 175: not a standard format. Is this a personal communication?

Line 177: why the ISMN is unlike the NASMD? Please elaborate

Line 181-182? Be specific on what parameters are used, or at least mention the list in Table S1.

Line 198: how to harmonize? Not clear

Line 199: where is section 2.3?

Line 214: what soil moisture map ? daily or annual? If it is daily, how did you do the aggregation?

Line 241-249. I don’t understand why a spatial interpolation for R2 map is needed. I don't think an interpolated R2 can be used for justification. Using an Ordinary Kriging doesn't make sense as Ordinary Kriging assumes your R2 points is spatially-correlated.

Supplement Figure 3A. There is no dotted blue line.

Table 1. The Unit of RMSE is needed.

6. PLOS authors have the option to publish the peer review history of their article (what does this mean?). If published, this will include your full peer review and any attached files.

Reviewer #1: No

Reviewer #2: No

---

## [Author Response · Author response to Decision Letter 0]

6 Sep 2019

Response to reviewers: PONE-D-19-17266

Downscaling Satellite Soil Moisture using Geomorphometry and Machine Learning

General journal comments

Comment: Thank you for submitting your manuscript to PLOS ONE. After careful consideration, we feel that it has merit but does not fully meet PLOS ONE’s publication criteria as it currently stands. Therefore, we invite you to submit a revised version of the manuscript that addresses the points raised during the review process.

Response: Thank you for the opportunity to submit a revised version of our work. We have revised the points raised by the reviewers and we believe that now this is a substantially improved manuscript. 

Comment: To enhance the reproducibility of your results, we recommend that if applicable you deposit your laboratory protocols in protocols.io, where a protocol can be assigned its own identifier (DOI) such that it can be cited independently in the future. For instructions see: http://journals.plos.org/plosone/s/submission-guidelines#loc-laboratory-protocols

Response: We have now included the protocol to reproduce this research in “protocols.io”. The link is dx.doi.org/10.17504/protocols.io.6cahase and is cited in the manuscript in line 283.

Comment: Thank you for stating the following in the Acknowledgments Section of your manuscript: "MG acknowledges a fellowship from CONACyT. RV acknowledges support from the National Science Foundation CIF21 DIBBs (Grant #1724843)."

"RV acknowledges support from the National Science Foundation CIF21 DIBBs (Grant #1724843). "

a. Please provide an amended Funding Statement that declares *all* the funding or sources of support received during this specific study (whether external or internal to your organization) as detailed online in our guide for authors at http://journals.plos.org/plosone/s/submit-now

a. Please state what role the funders took in the study. If any authors received a salary from any of your funders, please state which authors and which funder. If the funders had no role, please state: "The funders had no role in study design, data collection and analysis, decision to publish, or preparation of the manuscript."

Response: This section has been edited following the journal suggestions for acknowledgements. 

Comment: We note that you have stated that you will provide repository information for your data at acceptance. Should your manuscript be accepted for publication, we will hold it until you provide the relevant accession numbers or DOIs necessary to access your data. If you wish to make changes to your Data Availability statement, please describe these changes in your cover letter and we will update your Data Availability statement to reflect the information you provide.

Response: We have included the data repository information and obtained a DOI . 

Guevara, M., R. Vargas (2019). Annual soil moisture predictions across conterminous United States using remote sensing and terrain analysis across 1 km grids (1991-2016), HydroShare, https://doi.org/10.4211/hs.b8f6eae9d89241cf8b5904033460af61

Comment: Please include captions for your Supporting Information files at the end of your manuscript, and update any in-text citations to match accordingly. Please see our Supporting Information guidelines for more information: http://journals.plos.org/plosone/s/supporting-information

Response: We have included captions of the Supporting Information at the end of the revised manuscript. 

 

Additional Editor Comments:

Comment: I have received two reviews for your manuscript, both consider the work interesting and important for publication. One reviewer has several detailed questions that need to be addressed and will improve the manuscript's clarity and robustness. Consider the decision between and minor and major revision as the responses need to be addressed in detail.

Response: We would like to thank the two referees and editor for reviewing our manuscript, for providing constructive suggestions and for the opportunity to resubmit a revised and more detailed version of the manuscript. We have carefully reviewed the comments and have improved the manuscript accordingly.

 

Reviewers' comments:

Reviewer #1: 

Comment: I enjoyed reading the paper by Guevara et al., which addresses the downscaling of soil moisture data using geo-morphometry. The work is useful and very well explained in the manuscript. I recommend to accept it with few minor comments:

Response: We would like to thank Reviewer #1 for his/her valuable feedback to improve this work. We have revised the comments and improved the manuscript accordingly. 

Comment: Authors can add some references about different disaggregation methods applied on soil moisture datasets for downscaling, specifically related to topography or geomorphology, like: Pellenq et al 2003, A disaggregation scheme for soil moisture based on topography and soil depth

Response: As suggested by the reviewer, we have included new references of soil moisture disaggregation methods based on topography in introduction and discussion. The added references are (lines 111:113):

Pellenq J, Kalma J, Boulet G, Saulnier G-M, Wooldridge S, Kerr Y, et al. A disaggregation scheme for soil moisture based on topography and soil depth. Journal of Hydrology. 2003;276: 112–127. doi:10.1016/s0022-1694(03)00066-0

Busch FA, Niemann JD, Coleman M. Evaluation of an empirical orthogonal function-based method to downscale soil moisture patterns based on topographical attributes. Hydrological Processes. 2011;26: 2696–2709. doi:10.1002/hyp.8363

Comment: 2. Line 193: It will be nice to provide explicitly what are the primary hydrologically meaningful terrain parameters authors are referring to.

Response: We have improved the definition of primary and secondary terrain parameters in the introduction (lines 97-100) and methods (lines 210-216) of the revised version of the manuscript.

Comment: 3. A little description of these terrain parameters are required to be discussed.

Response: We have improved our description of these terrain parameters in the introduction (lines 97-100) and methods (lines 210-216). We have briefly discussed the main implications of using only these parameters for downscaling soil moisture in this revised version (see lines 435-437). 

Comment: 4. It will be great to add some discussion about the systematic error (Bias)

Response: We have included more information about systematic error (mean bias and RMSE) in our results section (lines 391-395, 500-511). Please note from Table 1 a consistent (cross-validated) error estimate (0.03 m3/m3) across all analyzed years. This error is larger when compared against the NASMD but similar between both products (0.057 for the downscaled estimate, and 0.062 for the original product). From our field validation, we also found a general negative mean bias in the original estimates (-0.051) that propagates to our downscaled product (-0.048). Note that systematic error (indicated by the mean bias and the RMSE is slightly lower than in the downscaled product), supporting the significant improvement in the explained variance (from 0.01 to 0.47) described in Figure 5 of the main text. 

 

Reviewer #2: 

Comment: The manuscript by Guevara and Vargas, entitled 'Downscaling Satellite Soil Moisture using Geomorphometry and Machine Learning', covers an interesting topic appropriate for PLOS ONE. The authors developed an approach to downscale the annual soil moisture at 25 km resolution derived from ESA CCI satellite products to 1 km resolution using a machine learning approach. A set of geomorphological parameters were applied to train the model using a k-NN algorithm. I think the study is original in the sense that it provides an alternative approach for downscaling methodologies. However, I feel the downscaled results were not well justified and the writing are not clear and can be strengthened. I have several concerns as listed below:

Response: We would like to thank Reviewer #2 for his/her comments. We have revised the concerns and clarified the text to show how our results support our main conclusion. 

Comment: The title is a bit misleading as it's too generalized. The current title doesn't suggest for what temporal resolution (annual or monthly), nor for what regions this approach can be applied. Validating the results for the US continent doesn't necessarily prove that it can be applied in other regions that have different climate (e.g. high latitudes and tropics).

Response: This manuscript presents a prove of concept for the application of terrain parameters for downscaling satellite derived soil moisture. Although the region of interest does not represent all the variability encountered across the world, it does represent a large climatic and ecological gradient. Our results show that indeed the accuracy of the method will vary across areas with different data availability for training and validating models (e.g., areas of large gaps in the ESA-CCI soil moisture product). We focus our validation framework in the US given the relatively high availability of soil moisture field data (and relatively good coverage of the ESA-CCI soil moisture product), which is usually sparse or not existent across large areas in higher latitudes or in the tropics. The opportunity for validation is the main reason why we chose this region but we are working on a larger application of this method across the world (manuscript in preparation).

Comment: The calculation of terrain parameters. It is surprised to see that some of the parameters in Figure 2 show fairly low values with almost none spatial heterogeneity. Is that a rendering problem or due to the normalization? This seems incorrect. For example, the wetness index is supposed to have significant spatial heterogeneity. See HYDRO1K or HydroSHEDS products for some of these parameters.

Response: The reviewer’s concern is mainly an issue of visualization of Figure 2, but the high resolution products at 1km resolution are available at https://www.hydroshare.org/resource/b8f6eae9d89241cf8b5904033460af61/. 

To show this we present in this response to the reviewers an example of the spatial detail in the topographic wetness index using 1 km grids. We also clarify that there are contrasting ranges of values in the terrain parameters calculated across CONUS (radians, meters, unitless). Consequently, we normalized these parameters by centering their means to 0 by a unit of variance. That is the reason why there are apparent low levels of variation in some parameters of Figure 2 in the main text. The figure below is showing a screenshot of the spatial heterogeneity (at the national level and across an elevation gradient of the Appalachian mountains) of the topographic wetness index at 1km pixel size, used in this study. Again, the final products/maps at 1km resolution are now available at https://www.hydroshare.org/resource/b8f6eae9d89241cf8b5904033460af61/.

FIGURE legend. Example of spatial detail of the topographic wetness index using 1km grids. a) topographic wetness index at 1km of spatial resolution across the conterminous United States (screenshot from SAGA-GIS), b) zoom in to the topographic wetness index across an elevation gradient in the Appalachian mountains of eastern CONUS. 

Comment: The downscaled 1k soil moisture. The results don’t suggest that it maintain the original values of the satellite product. Why the downscaled results seem to have a systematic lower value than the original one? Is that because of the calculation of terrain parameters? Also, the spatial heterogeneity is not higher in the downscaled product as it’s supposed to be. Downscaled products are supposed to provide more details at a higher resolution. The legend of Figure 3 and Figure S2 are not consistent, which makes it hard to compare the downscaled with the original visually.

Response: We clarify that our results show that the statistical distribution of the downscaled product is similar to the original ESA-CCI (see figure below). We also show that within the 95% confidence intervals (CI) our results capture relatively well the original variability of the ESA-CCI (i.e., no statistical differences between PDFs). We clarify that representing extreme values (i.e., >95% CI) is challenging for any model and our model is relatively lower for values >95% CI that represent the minority of the dataset. Most importantly, our results show that the downscaled soil moisture product has a better agreement with the “ground truth information” from the NASMD than the original ESA-CCI (see Figure 5), thus this demonstrates a substantial improvement using our proposed approach.

FIGURE legend. Probability distribution functions (PDFs) of the original satellite estimate (black) and the downscaled product (red). The vertical lines show the limits of the 95% of the data distribution of both soil moisture products. 

Comment: The parameters used in the machine learning. This study uses a few terrain parameters to train their model - Here is an implicit assumption that the soil moisture is mainly controlled by geomorphological information. But soil moisture is proved to be strongly influenced by vegetation via its regulation of evapotranspiration.

Response: The reviewer is correct to say that vegetation is also important to regulate soil moisture but we want to avoid including vegetation information into our model to have an independent estimate that can be used/compared with vegetation information in further analyzes. If vegetation information is included in model predictions then the resulting soil moisture could lead to spurious correlations in further soil moisture – vegetation analyses. This is why we kept an independent product that will avoid potential statistical redundancies by future users/analyzes.

Consequently, we focused our methods only on topographic parameters and our results show that there is an improvement in agreement with the NASMD compared with the original ESA-CCI product. Previous studies have also found a topographic signal on satellite soil moisture (Mason et al., 2016) with potential implications for downscaling satellite soil moisture estimates. Here, we show how combining (in a model) satellite soil moisture estimates and multiple terrain parameters is useful to generate a soil moisture product that showed higher agreement with field soil moisture measurements compared with the satellite soil moisture estimates only. Given the influence of topography on hydrological, climatological and ecological variability, our results support the reliability of the implicit assumption highlighted by the reviewer, when a) the input data is satellite soil moisture, b) the geomorphological information is represented by digital terrain parameters derived by the means of geomorphometry and c) the statistical model should be able to account for non linear relationships and it should be parameterized and validated with some resampling technique (such as repeated cross validation) or field soil moisture data (e.g., in the NASMD or the ISMN). 

Reference

Mason, D. C., Garcia-Pintado, J., Cloke, H. L. and Dance, S. L.: Evidence of a topographic signal in surface soil moisture derived from ENVISAT ASAR wide swath data, International Journal of Applied Earth Observation and Geoinformation, 45, 178–186, doi:10.1016/j.jag.2015.02.004, 2016.

Specific comments:

Comment: Paragraph Line 58-68. It seems the first outline sentence doesn’t match the following content.

Response: The connection of ideas in this paragraph has been improved in the revised version of the manuscript. 

Comment: Line 164-165: need to be more specific on how you did the downscale.

Response: We have improved the narrative of this sentence explaining the main components of our downscaling effort. We have also made available our protocol for downscaling satellite soil moisture using topography for improving the transparency and reproducibility of this research: dx.doi.org/10.17504/protocols.io.6cahase. 

Comment: Line 175: not a standard format. Is this a personal communication?

Response: It is a research manuscript, we have edited the citation style.

Comment: Line 177: why the ISMN is unlike the NASMD? Please elaborate

Response: We meant to say that the standardization procedures are different between the global effort and the North American effort. The differences between the NASMD and the ISMN are in the quality control and are explained and discussed in previous studies (Dirmeyer et al., 2016). While the ISMN uses global reference values, the NASMD uses a quality control based on North America reference values and the NASMD includes the information contained in the ISMN for CONUS. In the ISMN ‘basic quality control is performed and records suspected to be out-of-range or otherwise untrustworthy are flagged, but no data is omitted’ (Dirmeyer et al., 2016). The NASMD provides a single (quality controlled) value for each station (location, i.e., latitude and longitude) measuring soil moisture, while the ISMN provides raw data from each soil moisture instrument on each station (location, i.e., latitude and longitude) measuring soil moisture. This the NASMD integrates a single value from multiple soil moisture instruments taking measurements simultaneously and this more convenient that using the raw values of each instrument for the purpose of comparing values with pixels of soil moisture that are representative of a larger area than the soil moisture field station. 

Reference 

Dirmeyer, P. A., Wu, J., Norton, H. E., Dorigo, W. A., Quiring, S. M., Ford, T. W., et al. (2016). Confronting Weather and Climate Models with Observational Data from Soil Moisture Networks over the United States. Journal of Hydrometeorology, 17(4), 1049–1067. https://doi.org/10.1175/jhm-d-15-0196.1

Comment: Line 181-182? Be specific on what parameters are used, or at least mention the list in Table S1.

Response: Now we clearly explain which parameters are used and we explicitly refer the reader to Table S1 (line 193 and line 217).

Comment: Line 198: how to harmonize? Not clear

Response: We explain that we extracted the values of terrain parameters at the locations of the central coordinates of pixels in the coarser gridded soil moisture product. (lines 217-219)

Comment: Line 199: where is section 2.3?

Response: This was a mistake, we have edited the sentence. 

Comment: Line 214: what soil moisture map ? daily or annual? If it is daily, how did you do the aggregation?

Response: We meant to say that a model was built for each year of satellite soil moisture data. These satellite data was aggregated from its original temporal resolution to a yearly basis based on the median value. The time scale of our predictions are clearly explained in the abstract and throughout the manuscript.

Comment: Line 241-249. I don’t understand why a spatial interpolation for R2 map is needed. I don't think an interpolated R2 can be used for justification. Using an Ordinary Kriging doesn't make sense as Ordinary Kriging assumes your R2 points is spatially-correlated.

Response: We think that the R2 map provides information on the accuracy of our framework across areas where no field data for validation is available. We provide the R2 values (i.e., model results with the NASMD) in Figure 6. Then, we provide Figure 4f to show the spatial trend of the R2 values across CONUS. This map is supported by a variogram analysis (showing evidence of R2 points spatially-correlated) and we report parameter values (sill > nugget) in the methods section (lines 277-279). These parameters and the fitted variogram model are shown along the R2 Kriging prediction and its associated Kriging standard error in the figure below (see figure below). 

FIGURE 3. Screenshot of the native plot of the fitted R2 Kriging model (in the R package automap). This screenshot includes the R2 Kriging prediction and its associated Kriging standard error. The variogram parameters and the fitted variogram model is included in this screenshot. 

Comment: Supplement Figure 3A. There is no dotted blue line.

Response: This was an issue with the original resolution of the figure. The blue line indicates soil moisture in Supplementary Figure 3B. The Supplementary Figure S3A shows the cloud of points only, and the Figure S3B shows the variables. We have highlighted the soil moisture blue line to facilitate the interpretation of Supplementary Figure S3B. 

Comment: Table 1. The Unit of RMSE is needed.

Response: We have included the units of the RMSE in Table 1.

---

## [Editor Report · Decision Letter 1]

10 Sep 2019

Downscaling Satellite Soil Moisture using Geomorphometry and Machine Learning

PONE-D-19-17266R1

Dear Dr. Vargas,

We are pleased to inform you that your manuscript has been judged scientifically suitable for publication and will be formally accepted for publication once it complies with all outstanding technical requirements.

With kind regards,

Benjamin Poulter

Academic Editor

PLOS ONE

Additional Editor Comments (optional):

Thank you for your re-submission to PLOS ONE and for addressing the reviewer comments in detail. I am pleased to accept this version for publication.
---

## [Editor Report · Acceptance letter]

16 Sep 2019

PONE-D-19-17266R1 

Downscaling Satellite Soil Moisture using Geomorphometry and Machine Learning 

Dear Dr. Vargas:

I am pleased to inform you that your manuscript has been deemed suitable for publication in PLOS ONE. Congratulations! Your manuscript is now with our production department. 

With kind regards,

on behalf of

Dr. Benjamin Poulter 

Academic Editor

PLOS ONE